# The effect of climate change on yellow fever disease burden in Africa

Katy AM Gaythorpe[1]*, Arran Hamlet[1], Laurence Cibrelus[2], Tini Garske[1], Neil M Ferguson[1]

[1]Imperial College London, London, United Kingdom; [2]World Health Organisation, Geneva, Switzerland

**Abstract** Yellow Fever (YF) is an arbovirus endemic in tropical regions of South America and Africa and it is estimated to cause 78,000 deaths a year in Africa alone. Climate change may have substantial effects on the transmission of YF and we present the first analysis of the potential impact on disease burden. We extend an existing model of YF transmission to account for rainfall and a temperature suitability index and project transmission intensity across the African endemic region in the context of four climate change scenarios. We use these transmission projections to assess the change in burden in 2050 and 2070. We find disease burden changes heterogeneously across the region. In the least severe scenario, we find a 93.0%[95%CI(92.7, 93.2%)] chance that annual deaths will increase in 2050. This change in epidemiology will complicate future control efforts. Thus, we may need to consider the effect of changing climatic variables on future intervention strategies.

## Introduction

Yellow Fever (YF) is a vaccine preventable, zoonotic, arbovirus endemic in tropical regions of Africa and Latin America. It is responsible for approximately 78,000 deaths per year, although under reporting is high and since YF has a non-specific symptom set, misdiagnosis is an issue (*Garske et al., 2014*). YF has three transmission 'cycles' in Africa: urban, zoonotic and intermediate. The urban cycle, mediated by *Aedes Aegypti* mosquitoes, is responsible for explosive outbreaks such as the one seen in Angola in 2016 (*Ingelbeen et al., 2018*; *Wilder-Smith and Monath, 2017*).

While the urban cycle can rapidly amplify transmission, the majority of YF infections are thought to occur as a result of zoonotic spillover from the sylvatic reservoir in non-human primates (NHP). This zoonotic cycle is mediated by a variety of mosquito vectors including *Aedes africanus* and, as the NHP hosts are mostly unaffected by the infection in Africa, the force of infection due to spillover is fairly constant, although land use change has been shown to affect this (*Monath and Vasconcelos, 2015*). The intermediate cycle is sometimes called the savannah cycle and is mediated by mosquitoes such as *Ae. luteocephalus*, who feed opportunistically on humans and NHP, although human-human transmission is limited (*Barrett and Higgs, 2007*).

The Intergovernmental Panel on Climate Change (IPCC) states that global mean temperatures are likely to rise by 1.5°C, compared with pre-industrial levels, by between 2030 and 2052 if current trends continue (*Masson-Delmotte et al., 2018*). Increases are projected not only in mean temperature but also in the extremes of temperature, extremes of precipitation and the probability of drought (*Kharin et al., 2013*; *Dunning et al., 2018*).

With multiple mosquito vectors and a zoonotic cycle depending on NHP hosts, the impact of climate change on YF is likely to be complex. Focusing on the main urban vector, *A. aegypti*, there is strong evidence that projected climate change will alter its global distribution and thus, the risk of diseases it carries (*Ryan et al., 2019*; *World Health Organisation, 2018*; *World Health Organisation, 2018*). Climate change has been predicted to increase the regions at risk from dengue and

*For correspondence:
k.gaythorpe@imperial.ac.uk

Zika transmission, although seasonal variation in temperature may mitigate the likelihood of outbreaks in areas at the edges of the endemic zone (*Mordecai et al., 2017*; *Huber et al., 2018*).

Long-term projections of the future disease burden of YF are needed to inform vaccination planning (*VIMC, 2019*). Furthermore, differences due to climate change may increase the risk of epidemics, a key consideration for the Eliminate YF Epidemics (EYE) strategy (*World Health Organization, 2017*).

In this manuscript, we extend an existing model of YF occurrence and disease burden to incorporate a nonlinear temperature suitability metric (*Garske et al., 2014*). We estimate temperature suitability for YF based on the thermal response of the urban vector, *Ae. aegypti*, and the YF virus. We combine this with YF occurrence data in a Bayesian hierarchical model in order to account for uncertainty at each stage of the modelling process. This, along with established estimates of transmission intensity informed by serological survey data, allow us to predict current and future transmission intensity. Finally, we use ensemble climate model predictions of future temperature and precipitation to project transmission and thus, burden in 2050 and 2070. Our results are the first examination of YF burden under the potential future effect of climate change.

## Results

As we estimate a static force of infection, we focus on transmission as a result of sylvatic spillover rather than including the urban transmission cycle explicitly. As such, the results can be considered the estimated effect of climate change on sylvatic transmission and resulting burden.

### Model predictions for baseline scenario

*Figure 1* (left) shows occurrence of YF across Africa from 1984 to 2018. Incidence is focused in the West of Africa and, more recently, Angola and the Democratic Republic of the Congo. The model predicts a high probability of YF report in these areas and reflects the general patterns of YF occurrence, see *Figure 1* for comparison. Model fit can be characterised by the Area Under the Curve (AUC) statistic (*Huang and Ling, 2005*), which was 0.9004, similar to the original model formulation of *Garske et al., 2014*.

The predicted probability of a YF report is positively informed by temperature suitability with the median posterior predicted distribution shown in *Figure 2* (left). This highlights the high suitability of countries such as Nigeria and South Sudan for YF transmission. In contrast, Rwanda, Burundi and areas of Mali and Mauritania have low average temperature suitability. The fit of the thermal response models is shown in *Figure 2—figure supplements 1–4*.

### Projected transmission intensity

*Figure 2* (right) shows the median posterior predicted estimates of the force of infection for the baseline/current scenario, a comparison of the force of infection estimated only from serological studies, and those estimated from the GLM is provided in *Figure 2—figure supplement 1*. When we incorporate the ensemble projections of temperature and precipitation change we see heterogeneous impacts on force of infection. *Figure 3* shows the percentage change in median force of infection for the year 2070. Projections for 2050 are shown in *Figure 3—figure supplement 1*.

The posterior distributions of predicted changes in force of infection in different African regions are shown in *Figure 4* (region definitions shown in *Figure 4—figure supplement 1*). Projections for individual countries are given in the Appedix. In West Africa, the predicted change is clustered around zero in the majority of scenarios; this is particularly the case for year 2050. However, due to wider uncertainty in 2070 and for RCP scenario 8.5 in general, there is a more discernible increase. In the East and Central regions, a predicted increase in force of infection is more apparent. Whilst the differences between 2050 and 2070 are difficult to see for RCP scenario 2.6, both peak above zero. In RCP scenarios 4.5, 6.0 and 8.5, the distinction between years is clear, particularly in 8.5, with the greatest increases seen in 2070 as temperatures are expected to continue to rise.

When we examine the changes at country level, shown in the appendix, the changes are more heterogeneous. For RCP 2.6 Guinea Bissau, the change in force of infection in 2070 is potentially broad, with a credible interval spanning zero: 10.3% (95%CrI [−33.2% , 96.3%]). Whereas in Central African Republic, there is a notable increase by 87.1% (95%CrI [12.4% , 390.2%]).

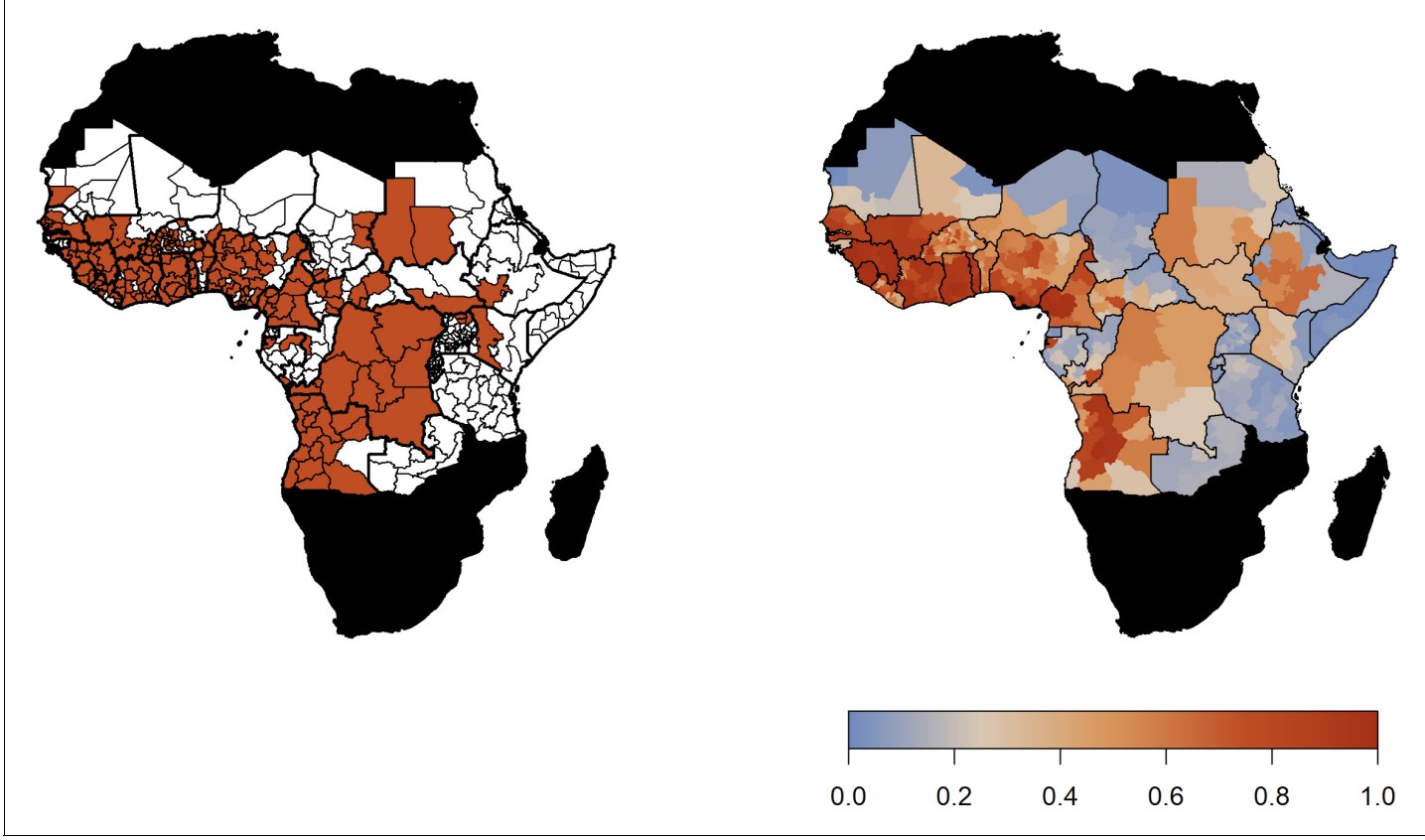

**Figure 1.** Observed YF occurrence (left) and median probability of a YF report predicted by the GLM.

The online version of this article includes the following figure supplement(s) for figure 1:

**Figure supplement 1.** Schematic of data sources and models adapted from *Gaythorpe et al., 2019*.

**Figure supplement 2.** Comparison of force of infection estimates for each admin level 1 unit where we have serological surveys between the estimate from serological surveys only and the GLM within the Bayesian hiearchical model.

**Figure supplement 3.** Median posterior predicted deaths in 2050 (log10 scale).

**Figure supplement 4.** Median posterior predicted deaths in 2070 (log10 scale).

## Projected burden

The projected percentage change in the annual number of deaths caused by YF across Africa is given in *Table 1*; the projected annual deaths per capita for endemic countries are shown in *Figure 5* and in *Figure 5—figure supplement 1*. These projections assume vaccination is static from 2019 onwards that is that only routine vaccination continues at 2018 levels. Similarly, we assume case management is unvarying. Aggregated numbers of deaths per country and region are shown in the appendix.

While lower 95% credible intervals in *Table 1* are negative, the overall posterior probabilities that climate change will increase YF mortality are very high for each climate scenario. The probability that deaths will increase is 95.5% (95% CrI [95.3%, 95.7%]) for RCP 2.6 in year 2070, rising to 95.9% (95% CrI [95.7%, 96.1%]) for RCP 8.5 in year 2070, values for all scenarios and years are shown in appendix 1.

As with the force of infection projections, the most severe increases are seen for RCP scenario 8.5, especially in year 2070. The distinction between current projected deaths per capita and those under each RCP scenario are most clearly seen for countries in Central Africa, such as Central African Republic, and East Africa, such as Ethiopia. The four countries with the least distinct change, Liberia, Guinea, Sierra Leone and the Gambia, are all in West Africa, commonly thought to see the most intense YF transmission. As such, it appears that the most marked increases in burden are found in East and central Africa.

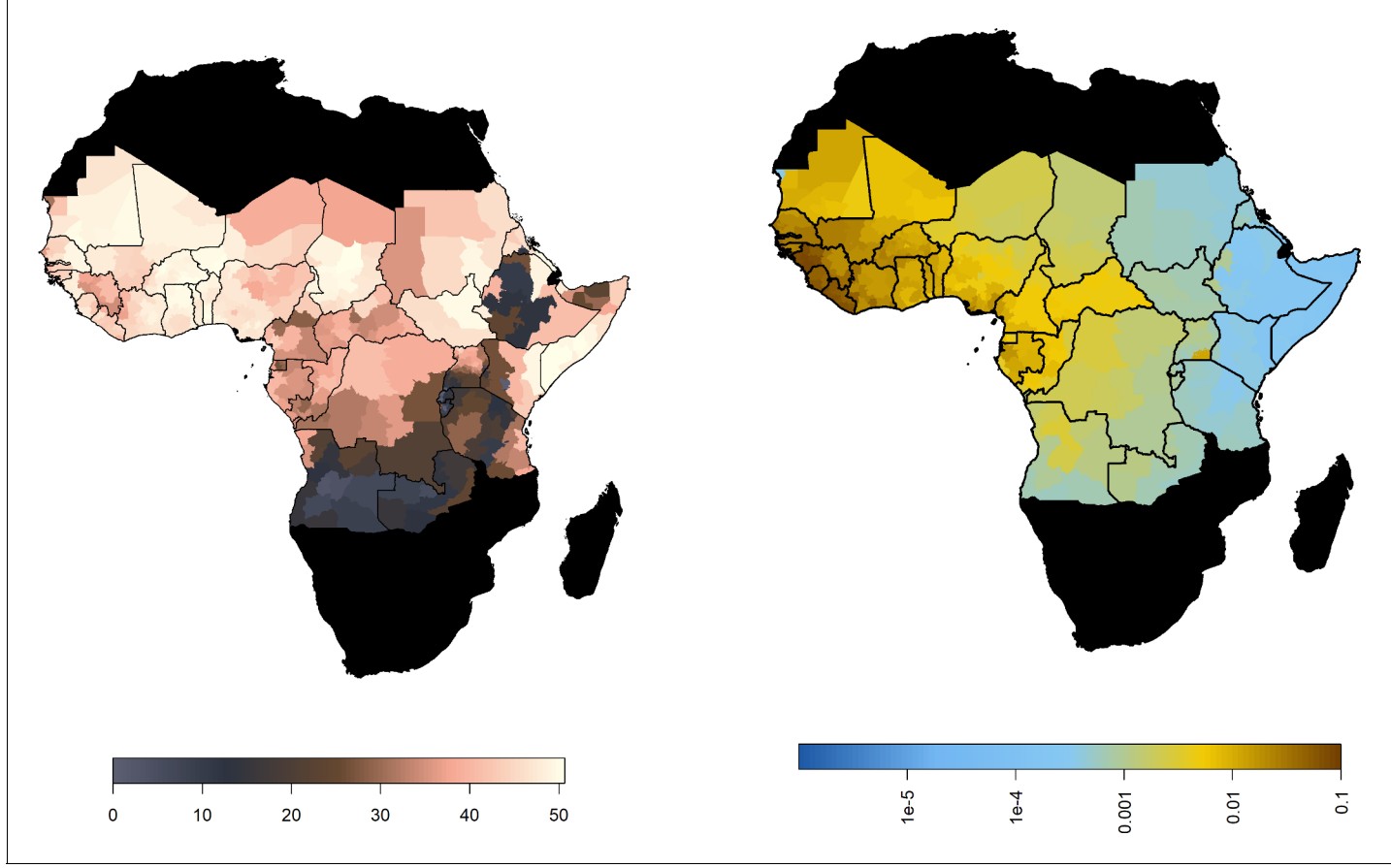

**Figure 2.** Median predicted model outputs for baseline scenario. (Left) Median posterior predicted temperature suitability for the African endemic region with average temperature. (Right) Median predicted FOI for the African endemic region at baseline.

The online version of this article includes the following figure supplement(s) for figure 2:

**Figure supplement 1.** Bite rate per day of *Aedes aegypti* mosquitos in response to temperature change.

**Figure supplement 2.** Mortality rate per day of *Aedes aegypti* mosquitos in response to temperature change.

**Figure supplement 3.** Inverse extrinsic incubation period in response to temperature change.

**Figure supplement 4.** Temperature suitability in response to temperature change.

## Discussion

We build on an established model of YF occurrence and transmission to accommodate temperature and precipitation projections for four climate emissions scenarios. Non-linear dependence on temperature was incorporated by utilising a function of temperature suitability, informed by thermal response data for *A. aegypti*. We jointly estimated parameters for the temperature suitability and occurrence models in a Bayesian framework, allowing us to quantify the uncertainty in our projections. We found that model fit remained good with a median AUC of 0.9004 despite necessary changes to the covariates used in the occurrence model compared with past work *Garske et al., 2014*; where changes were required in order to include covariates for which climate change projections were available. This gave us some confidence in the suitability of the model for projecting the impact of climate change on YF transmission through to 2070, the last year for which climate emission scenario projections are available for temperature and precipitation.

The force of infection is projected to increase for the majority of countries in each scenario. Consistently, the Central African Republic is one of the countries most likely to see an increase in transmission, while Liberia and Guinea Bissau have more uncertain projections. This highlights that the most severe proportional increases in force of infection are seen outside West Africa. However, as transmission is currently highest in West Africa, even a small future relative increase of 3% (seen for

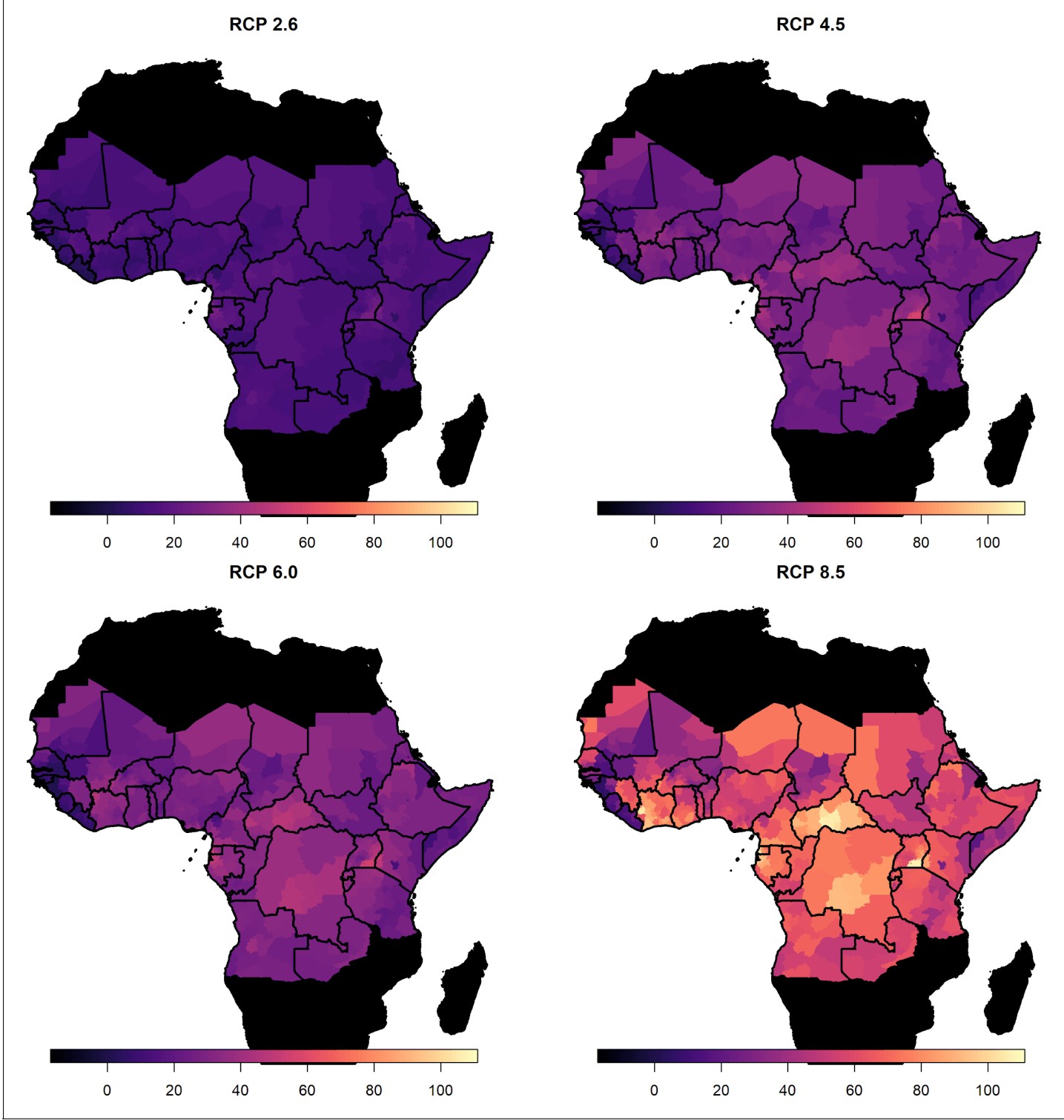

**Figure 3.** Percentage change in force of infection in 2070. Median predicted change in force of infection in the African endemic region in 2070 for the four emission scenarios.

The online version of this article includes the following figure supplement(s) for figure 3:

**Figure supplement 1.** Percentage change in force of infection in 2050.

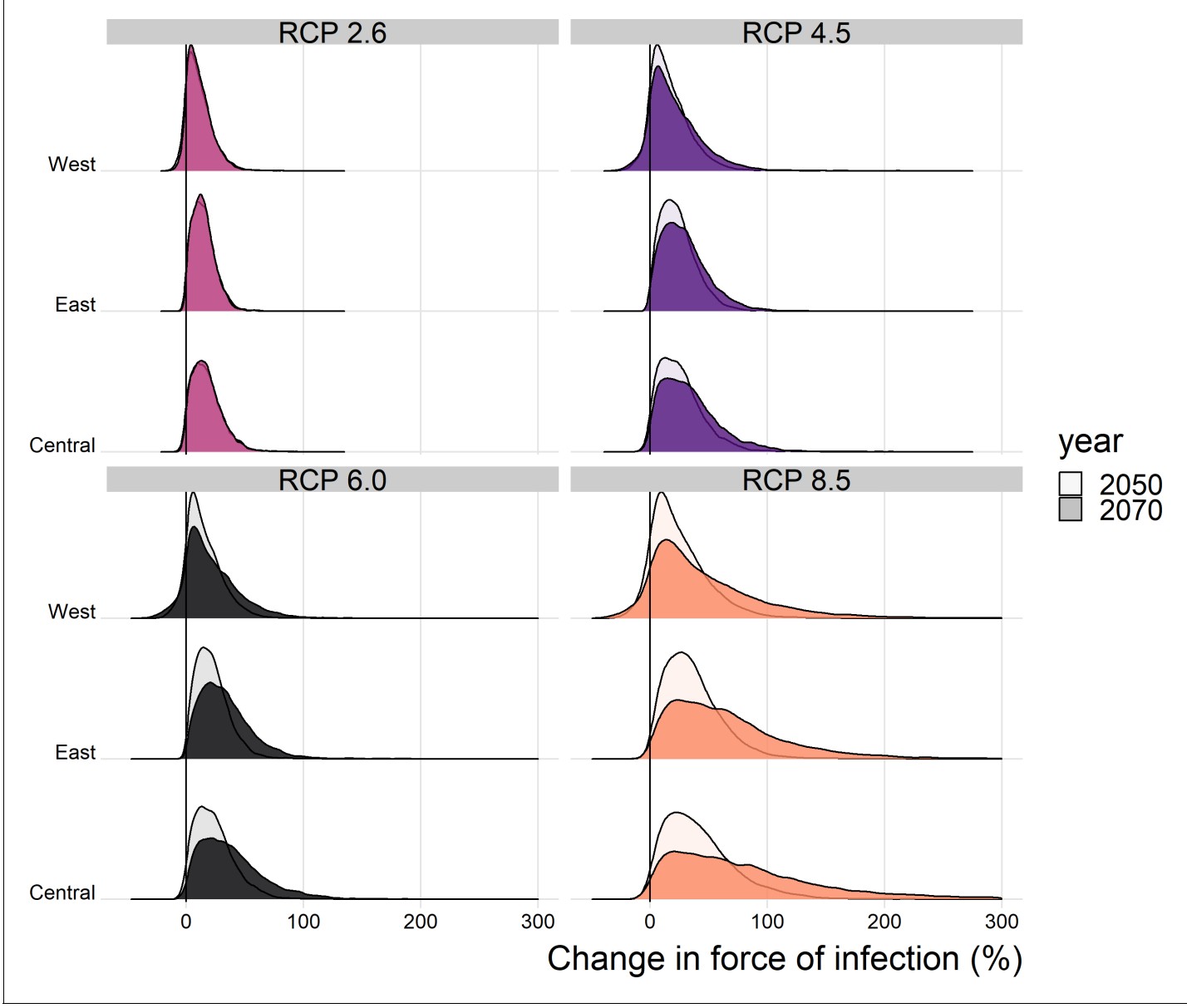

**Figure 4.** Posterior distribution of the change in the spatial mean force of infection (%) for each region of Africa, year and climate scenario. The online version of this article includes the following figure supplement(s) for figure 4:

**Figure supplement 1.** Country groupings in regions used in the manuscript with West, yellow; Central, purple and East, green.

Liberia in scenario RCP 2.6 in year 2050, see appendix) could equate to a substantial increase in the projected absolute number of annual YF deaths.

In all scenarios, there is a high probability that the number of deaths and deaths per capita will increase in the African endemic region. The most marked changes are seen for RCP 8.5, the most severe emission scenario; however, changes are heterogeneous geographically with large proportional increases occurring in Central and East Africa. We expect the number of deaths per year to increase by 10.0% (95% CrI [−0.7, 34.1]) under RCP scenario 2.6 or 40.0% (95% CrI [−2.9, 178.6]) under RCP scenario 8.5 by 2070 (see *Table 1* for other values).

We assume that the force of infection changes linearly between 2018 and 2050, and between 2050 and 2070. *Video 1* illustrates this by showing posterior samples of the change in deaths by region for all years between 2018 and 2070. For RCP scenario 2.6, deaths largely cease increasing after year 2050, in line with the assumption that RCP 2.6 represents the situation where contributing

**Table 1.** Predicted percentage change in deaths in the African endemic region in 2050 and 2070 compared to the baseline/current scenario.

| Year | Scenario | 95% CrI low | 50% CrI low | Median | 50% CrI high | 95% CrI high |
|------|----------|-------------|-------------|--------|--------------|--------------|
| 2050 | RCP 2.6  | −2.36 | 4.49  | 10.84 | 18.58 | 37.91  |
| 2050 | RCP 4.5  | −2.40 | 7.32  | 16.71 | 28.16 | 57.43  |
| 2050 | RCP 6.0  | −2.78 | 6.79  | 15.49 | 25.86 | 51.85  |
| 2050 | RCP 8.5  | −2.17 | 11.03 | 24.92 | 41.84 | 88.33  |
| 2070 | RCP 2.6  | −0.74 | 4.11  | 9.99  | 17.03 | 34.10  |
| 2070 | RCP 4.5  | −2.76 | 7.77  | 19.28 | 33.56 | 71.08  |
| 2070 | RCP 6.0  | −4.56 | 8.63  | 21.35 | 36.70 | 77.70  |
| 2070 | RCP 8.5  | −2.90 | 16.08 | 39.57 | 72.43 | 178.63 |

carbon activities peak by 2030; however, this scenario has been suggested to be 'unfeasible' (*Mora et al., 2013*; *van Vliet et al., 2009*). In RCP scenario 8.5, carbon contribution activities are assumed to continue increasing throughout the century. A potential impact of this is seen in the number of YF deaths predicted by our model in East and Central Africa, which accelerate after 2050.

Climate change may affect not only the magnitude of YF disease burden but also its distribution. We find that, through the projected changes in both temperature and rainfall, transmission may change heterogeneously across the region. This is emphasised by their individual contribution; in the appendix, we explore the effects of changes in only temperature or rainfall. This illustrates that whilst temperature change will drive the variation in transmission intensity with rainfall often acting to mitigate, in some countries there can be a 'perfect storm' of altering rainfall and temperature leading to increases in transmission that would not occur if only temperature was varying. This may lead to changing priorities with respect to vaccination. However, it is unclear whether the comparatively low proportional increase in burden seen for West Africa is due to more intensive vaccination or due to the limited increase in force of infection. Our results suggest that there could be drastic proportional increases in burden in East and Central Africa that may lead to greater vaccine demand in areas which have previously been of lower risk. Thus, whilst the countries experiencing the highest numbers of deaths will remain high risk, see *Figure 1—figure supplement 3* and *Figure 1—figure supplement 4* for the median distribution of deaths per year, countries such as Ethiopia and Somalia may become higher priority targets for vaccination.

Our analysis has a number of limitations. In order to utilise emission scenario projections, we were limited to covariates with projections in 2050 and 2070, namely temperature and precipitation. This meant that we adapted our previous best-fit model (*Garske et al., 2014*) to include temperature range, temperature suitability and precipitation rather than enhanced vegetation and landcover. This change slightly reduced fit quality, giving an AUC of 0.9004 as opposed to to 0.9157 (*Gaythorpe et al., 2019*). Vegetation is a key factor determining habitat of non-human primates, an element that may not be captured by the temperature suitability index which focuses on the vector *A. aegypti*. This omission may lead to an overestimation of the future burden as elements such as desertification and the impact of increasing frequencies of forest fires are not considered (*Overpeck et al., 1990*; *Huang et al., 2016*; *James et al., 2013*).

Similarly, whilst the RCP scenarios model socio-economic and land-use changes, we do not explicitly include these aspects here (*van Vuuren et al., 2011*). As such, we omit the human choices that may affect population distributions and behaviour, for example urbanisation which has been shown to both reduce disease burden (*Wood et al., 2017*) and increase emergence of arboviruses (*Gubler, 2011*; *Hotez, 2017*). In the same way, while our model accounts for migration through use of the UN WPP population data, climate scenario-specific migration is not included in the model. This may mean that we under estimate the potential increases in burden due to increased infringing of human environments on the sylvatic cycle. Projecting these non-linear relationships between human behaviour and transmission would be highly uncertain and is a source of ongoing research.

Vaccination is the main control method for yellow fever and whilst we account for vaccine coverage and efficacy in this manscript, we do not explicitly propagate uncertainty in vaccination

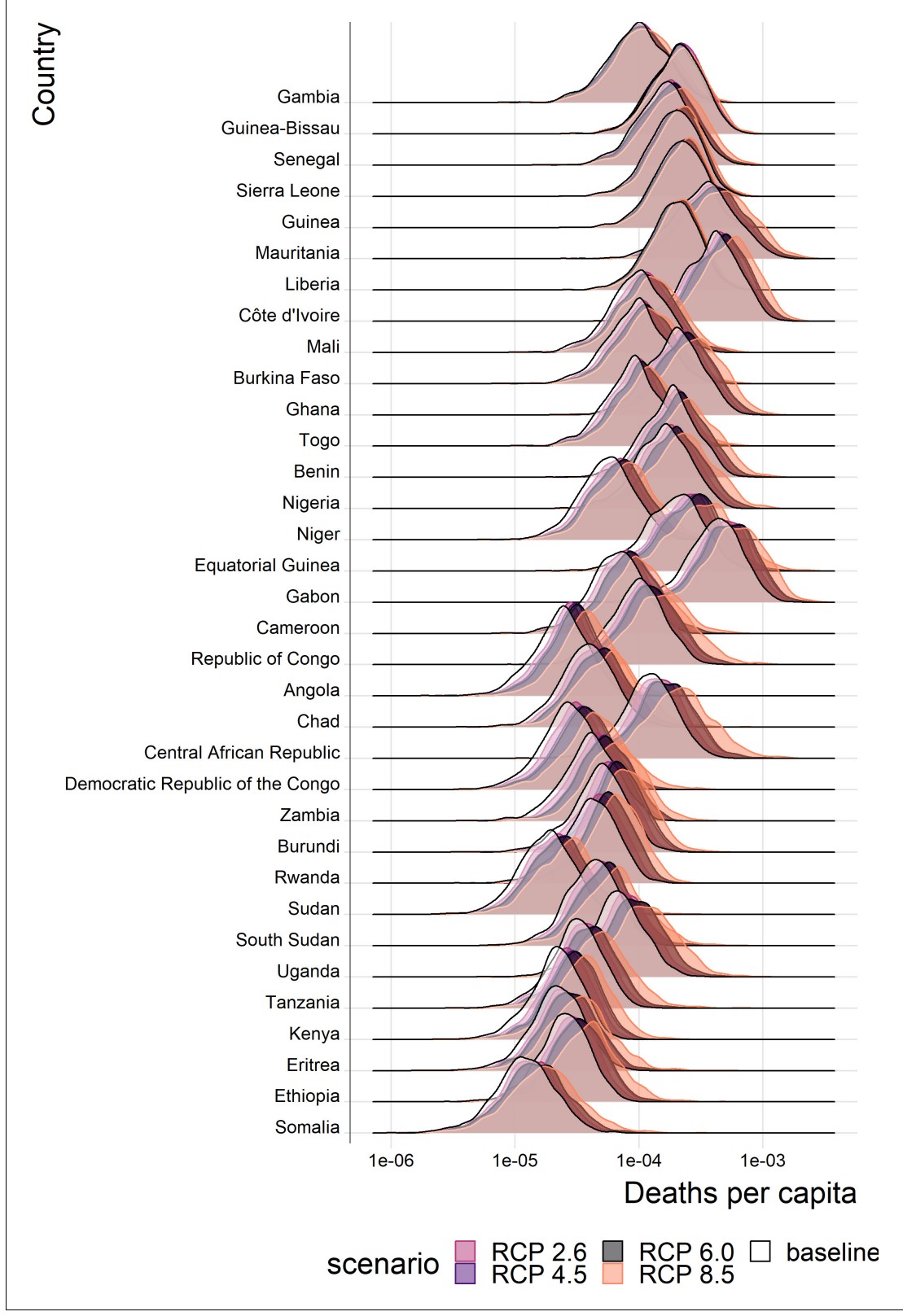

**Figure 5.** Posterior predicted annual YF deaths per capita for each country in the African endemic region in 2070. Countries are ordered by longitude. The online version of this article includes the following figure supplement(s) for figure 5:

**Figure supplement 1.** Posterior predicted deaths per capita for each country in the African endemic region in 2050.

*Figure 5 continued on next page*

coverage. This will be uncertain not only through data scarcity on vaccination campaign doses, wastage and clustering of doses, but also through the uncertainty in demography. We have presented a comparison of scenarios where, in all cases, vaccination coverage distribution, is held to be the same. As such, whilst we focus on the effect of changing transmission, we will underestimate the uncertainty in our estimates of burden in the future.

Data availability constrains aspects of our modelling approach. We use *A. aegypti* and YF-specific datasets to inform the thermal response relationships and thus, temperature suitability index. However, some data, such as information on the extrinsic incubation period are severely limited; we use a dataset of experimental results from 1930s (*Davis, 1932*). These data may be outdated due to current mosquito species potentially adapting to different climates as well as improved experimental procedures. This is a key data gap for YF and new experimental results concerning the extrinsic incubation period could provide valuable insight into the dynamics of the virus in mosquitoes today.

As further experimental data on thermal responses for *A. aegypti* and other vectors of YF become available, the temperature suitability index developed here will be able to be enhanced. YF is known to have multiple vectors, each contributing to transmission cycles differently (*Monath and Vasconcelos, 2015*), which are likely to have different thermal responses. Focusing only on the urban vector of YF, as we have in this manuscript, means that we will likely under-estimated the uncertainty in the thermal response of the vectors of YF and thus future projections of burden. Additionally, whilst we have included a relatively detailed relationship between transmission and temperature, we have only assumed a simple relationship with rainfall. Currently models of thermal response for vectors of diseases such as YF are well parametrised with experimental results; however, this is not yet the case for the influence of rainfall on transmission although there are clear links with aspects such as vector breeding. As these relationships are better characterised, we can further refine the relationships in the current work to reflect the more nuanced relationships between temperature, rainfall and transmission.

We focus only on a constant force of infection model which is similar to assuming the majority of transmission occurs as a result of zoonotic spillover. This assumption is supported by recent studies *Gaythorpe et al., 2019*; however, the urban transmission cycle, driven by *A. aegypti* plays a crucial role in YF risk and was responsible for recent severe outbreaks such as that in Angola in 2016. Incorporating climate projections into models that examine multiple transmission routes and thermal responses for multiple vectors, would produce a more realistic picture of how the dynamics of this disease may change with climate.

Climate change is projected to have major global impacts on disease distribution and burden (*Mordecai et al., 2017*; *Huber et al., 2018*; *Kraemer et al., 2015*). Here, we examined the specific effects on YF and find that disease burden and deaths are likely to increase heterogeneously across Africa. This emphasises the need to implement and prepare for new vaccination activities, and consolidate existing control strategies in order to mitigate the rising risk from YF. Intervention through vaccination is the gold standard for YF, and new approaches are being implemented with respect to fractional dosing which is a useful resort to respond to urban outbreaks in case of vaccine shortage

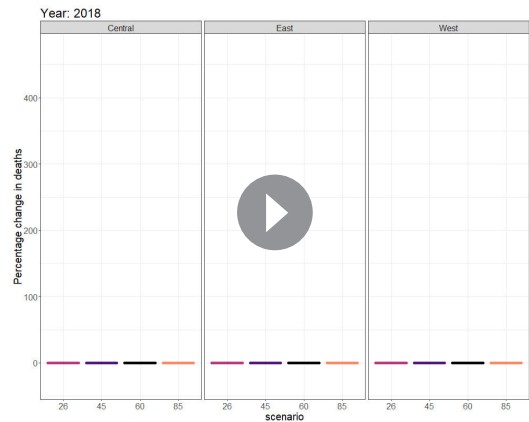

**Video 1.** Percentage change in deaths from 2020 to 2070 in three regions in the African Endemic region under 4 climate change scenarios. 100 samples of the posterior predicted trajectories are shown.
https://elifesciences.org/articles/55619#video1

(*Vannice et al., 2018*). Yet, vaccination is not the only potentially effective control for YF, with novel vector control measures such as the use of Wolbachia showing promise, and perspectives to improve clinical management or urban resilience (*Rocha et al., 2019*; *World Health Organization, 2017*). Finally, in order to monitor and respond to changing transmission patterns, effective and sensitive surveillance will be essential.

## Materials and methods

A schematic of data sources and models is shown in *Figure 1—figure supplement 1*.

### Datasets

We use a number of data sets to inform both the generalised linear model (GLM) of YF occurrence and the temperature suitability model. Additionally, we rely on estimates of transmission intensity informed by serological studies which are detailed in *Gaythorpe et al., 2019* and described below.

#### YF occurrence

Details of YF outbreaks occurring from 1984 to present day were collated into a database of occurrence, extended from *Garske et al., 2014*. These data were collected from the World Health Organisation (WHO) weekly epidemiological record (WER), disease outbreak news (DON), published literature and internal WHO reports (*World Health Organization, 2009*; *World Health Organization, 1996*). The database includes all outbreaks recorded for yellow fever and is resolved at province level, any reports that could not be resolved at province level were excluded. Additionally, reports of suspected YF cases were collected in the WHO African Regional Office YF surveillance database (YFSD); this included data from 21 countries in West and Central Africa. The database was based on the broad case definition of fever and jaundice leading to a large proportion of cases attributed to non-YF causes and cross-reactivity with other flaviviruses was not considered. However, the incidence of suspected cases can be used as a measure of surveillance effort and is included as a covariate in the generalised linear model. We assume this to be constant over time due to scarcity of data on the subject.

#### YF serological status

Surveys of seroprevalence were conducted in Central and East Africa. We use these to assess transmission intensity in specific regions of the African endemic zone. The current study includes surveys from published sources (*Diallo et al., 2014*; *Kuniholm et al., 2006*; *Merlin et al., 1986*; *Omilabu et al., 1990*; *Tsai et al., 1987*; *Werner and Huber, 1984*) and unpublished surveys from East African countries conducted between 2012 and 2015 as part of the YF risk assessment process (*Mengesha Tsegaye et al., 2018*). The surveys were included only if they represent the population at steady state, as such outbreak investigations were omitted (*Garske et al., 2014*). Additionally, in the majority of surveys, vaccinated individuals were not included; however, in South Cameroon, vaccination status is unclear and so we fit an additional vaccine factor for this survey. Summary details of the seroprevalence studies are included in the apendix.

#### Past vaccination coverage and demography

Vaccination coverage is estimated using data on historic large-scale mass vaccination activities taking place between 1940 and 1960 (*Durieux, 1956*; *Moreau et al., 1999*), routine infant immunisation reported by the WHO and UNICEF estimates of National Immunization Coverage (WUENIC) (*World Health Organization/ UNICEF, 2015*), outbreak response campaigns from 1970 onwards which are detailed in the WHO WER and DON (*World Health Organization, 2009*; *World Health Organization, 1996*) and recent preventive mass-vaccination campaigns carried out as part of the yellow fever initiative (*World Health Organisation, 2016*). The coverage is estimated with the methodology of Garske et al. and Hamlet et al. and is visualised in the polici shiny application (*Garske et al., 2014*; *Hamlet et al., 2018a*). The application provides vaccination coverage estimates at province level for 34 endemic countries in Africa which can be downloaded for years between 1940 and 2050. We assume all targeted age groups have an equal chance of vaccination irrespective of vaccination staus.

Demography is obtained from the UN World Population Prospect (UN WPP) (*DoE United Nations, 2017*). We dis-aggregate this to province level by combining it with estimates of spatial population distributions from LandScan 2014 (*Dobson et al., 2000*). This allows us to estimate population sizes at province level for each year of interest assuming that the age structure is relatively similar across all provinces in each country.

## Environmental and climate projections

We use three main environmental covariates within the generalised linear model of YF occurrence: mean annual rainfall, average temperature and temperature range, shown in *Figure 6* and listed in *Table 2*. These are gridded data at various resolutions, ranging from approximately 1 km to 10 km, which we average at the first administrative unit level (*Nasa LPD, 2001*; *Xie and Arkin, 1996*; *Hijmans et al., 2004*).

Projected temperature and rainfall changes under climate change scenarios were obtained from worldclim version 1.4 (*Hijmans et al., 2005*; *Fick and Hijmans, 2017*). These data provided the 5th Intergovernmental panel on climate change (IPPC5) climate projections for four Representative Concentration Pathways (RCPs): 2.6, 4.5, 6.0 and 8.5 (*van Vuuren et al., 2011*). The different RCPs indicate different possible emission scenarios and represent the resulting radiative forcing in 2100,

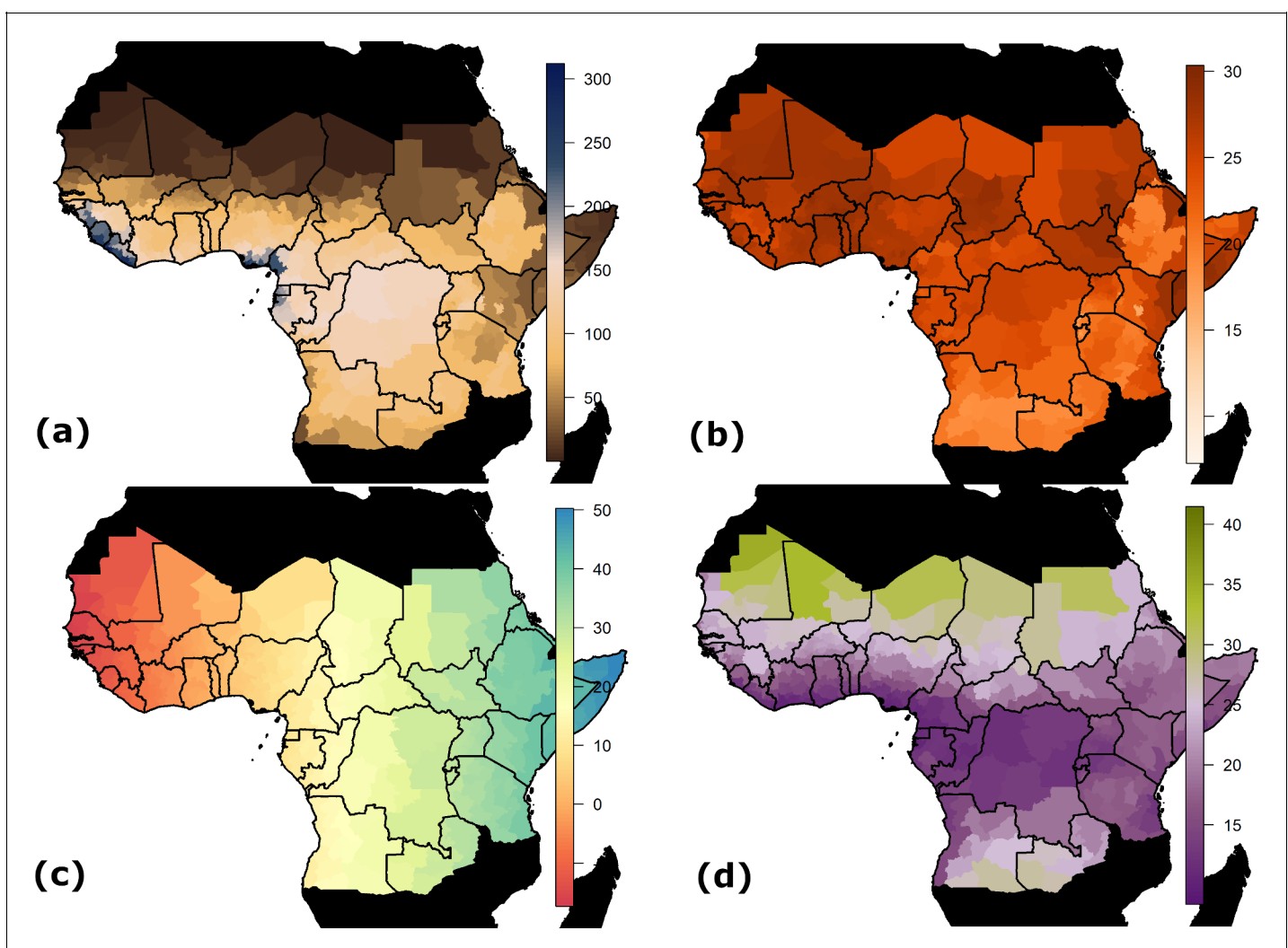

**Figure 6.** Spatial data inputs for generalised linear model. Countries shown in black are not considered endemic for YF. (a) Estimated mean monthly rainfall (mm) for baseline/current scenario. (b) Average temperature at baseline/current scenario in ℃. (c) Longitude. (d) Range in temperature at baseline/current scenario in ℃.

**Table 2.** Generalised linear model covariates.

| Covariate | Interpretation |
|---|---|
| log(survey quality) | Log of the survey quality for countries in YFSD. |
| adm05 | Country factors for countries not in YFSD. |
| longitude | Longitude of province centroid |
| temperature suitability | Temperature suitability at average suitability of province. |
| temperature range | Temperature range in province. |
| rainfall | Mean Precipitation in province. |
| log(pop) | Log of the human population size of the province |

measured in W/m$^2$ or watts per square metre, see *Table 3* for further information (*Stocker, 2013*). Each scenario is assumed to peak at a different times, with emissions peaking between 2010 and 2020 for RCP 2.6, but rising throughout the century for RCP 8.5. Projections of the mean global temperature rise by 2046–2065 are 1 or 2 °C for RCPs 2.6 or 8.5, respectively, compared to pre-industrial levels of the 1880s. By the end of the century, these projections suggest a rise of 1 [0.3 to 1.7] or 3.9 [2.6 to 4.8]°C for RCPs 2.6 or 8.5 (*Stocker, 2013*; *Rogelj et al., 2012*). Current warming is estimated to be 0.85 °C since pre-industrial levels (*Stocker, 2013*). Based on current commitments through aspects such as the Paris agreement, scenarios where temperatures are expected to rise by more than 3 °C have been suggested to be most likely (*Sanford et al., 2014*). As such, a recent study omitted the RCP 2.6 scenario as it is unlikely now to occur (*Mora et al., 2013*; *van Vliet et al., 2009*).

Projected mean rainfall, maximum temperature and minimum temperature are available for each RCP scenario in years 2050 and 2070. We take the midpoint and range of the temperature as inputs for the model of YF occurrence, where the midpoint temperature is used to calculate the temperature suitability index.

We do not model changes in climate prior to 2018, instead using Worldclim baseline estimates described as representative of conditions from 1960 to 1990 (*Hijmans et al., 2005*).

## Temperature suitability

We estimate the components of the temperature suitability index from YF-specific sources of information on extrinsic incubation period, vector mortality and bite rate for *A. aegypti*, the urban vector of YF (*Davis, 1932*; *Tesla et al., 2018*; *Hamlet et al., 2018b*; *Mordecai et al., 2017*). The extrinsic incubation period was estimated from the experimental results of Davis which were calculated specifically for YF in *A. aegypti* (*Davis, 1932*). We included bite rate data from both *Mordecai et al., 2017* and *Martens, 1998* which both describe *A. aegypti*. Finally, vector mortality was estimated from the experimental data of *Tesla et al., 2018*. Where data was provided in figure form, plots were digitised to extract the information. All data used for fitting the temperature suitability model are made available in the GitHub repo (https://github.com/mrc-ide/YF_climateChange; *Gaythorpe, 2020*; copy archived at https://github.com/elifesciences-publications/YF_climateChange). Whilst we focus only on thermal response of the urban vector of YF due to data availability, we estimate the thermal response models within a Bayesian hierarchical framework in order to capture some of the uncertainty that we miss from examining one vector species.

**Table 3.** Projected change in global mean surface air temperature and CO2 concentrations by 2100 relative to the reference period of 1986–2005 (*Stocker, 2013*).

| Scenario | Temperature rise (°C) [range] | CO2 concentrations (ppm) |
|---|---|---|
| RCP 2.6 | 1 [0.3 to 1.7] | 421 |
| RCP 4.5 | 1.8 [1.1 to 2.6] | 538 |
| RCP 6.0 | 2.2 [1.4 to 3.1] | 670 |
| RCP 8.5 | 3.7 [2.6 to 4.8] | 936 |

## Models

We reformulate an established model of YF occurrence to accommodate nonlinear dependence on temperature and rainfall (*Garske et al., 2014*; *Jean et al., 2020*; *Gaythorpe et al., 2019*). We couple this with established results from a transmission model of serological status to estimate transmission intensity across the African endemic region at baseline/current environmental conditions . Then, we project transmission intensity for four climate scenarios given projected changes in temperature and rainfall.

### YF occurrence

The generalised linear model (GLM) of YF occurrence provides the probability of a YF report at first administrative unit level for the African endemic region dependent on key climate variables. In order to assess the effect of climate change on YF transmission, we use the same methodology as (*Garske et al., 2014*; *Jean et al., 2020*; *Gaythorpe et al., 2019*); and incorporate covariates indicative of climate change that also have projections available in years 2050 and 2070 for different emission scenarios. As such, we omit enhanced vegetation index and land cover from the best fitting model of *Garske et al., 2014* in favour of the temperature suitability index which depends on the average temperature, the temperature range and average rainfall. Temperature and rainfall are known to have implications on both the vectors of YF and the distribution of the non-human primate reservoir (*Reinhold et al., 2018*; *Cowlishaw and Hacker, 1997*). However, the effect of temperature, particularly on vectors, is highly non-linear with increased mortality seen at very low and high temperatures; as such, we include the range in temperature as a covariate of our occurrence model as well as the non-linear temperature suitability index (*Mordecai et al., 2017*; *Tesla et al., 2018*). A full listing of covariates used in given in the appendix.

### Temperature suitability

We model suitability of the environment for YF transmission through temperature dependence. It has been shown that the characteristics of the virus and vector change with temperature (*Brady et al., 2014*; *Kraemer et al., 2015*; *Mordecai et al., 2017*; *Tjaden et al., 2018*). We model this using a function of temperature for the mosquito biting rate, the extrinsic incubation period and mortality rate for the mosquito which we combine to calculate the temperature suitability based on the Ross-MacDonald formula for the basic reproduction number of a mosquito-borne disease (*Macdonald, 1957*). In the below, we focus on *A. aegypti*.

The functional form used to model temperature suitability varies in the literature. We continue to use a form which can be parameterised solely from data specific to YF (*Hamlet et al., 2018b*; *Garske et al., 2013*). However, alternative formulations have been published in the context of other arboviral infections (*Mordecai et al., 2017*; *Ryan et al., 2019*; *Brady et al., 2014*; *Brady et al., 2013*; *Tjaden et al., 2018*).

Each input of the temperature suitability, $z(T)$, is modelled as a function of average temperature where the individual thermal response follow the forms of Mordecai et al. The temperature suitability equation is as follows:

$$z(T) = \frac{a(T)^2 \exp(-\mu(T)\rho(T))}{\mu(T)}, \tag{1}$$

where $T$ denotes mean temperature, $\rho$ is the extrinsic incubation period, $a$ is the bite rate and $\mu$ is the mosquito mortality rate. The thermal response models for $\rho$, $a$ and $\mu$ follow *Mordecai et al., 2017* as follows:

$$a(T) = a_c T(T - a_{T_0})(a_{T_m} - T)^{0.5},$$

$$\rho(T) = 1/\rho_c T(T - \rho_{T_0})(\rho_{T_m} - T)^{0.5},$$

$$\mu(T) = 1/(-\mu_c(T - \mu_{T_0})(\mu_{T_m} - T)),$$

where the subscripts $T_0$ and $T_m$ indicate respectively the minimum and maximum values of each

variable, and subscript $c$ labels the positive rate constant for each model. The three resulting parameters for each model are estimated by fitting to available experimental data. The mortality rate μ is limited to be positive.

## Mapping probability of occurence to force of infection

We utilise previously estimated models of seroprevalence informed by serological survey data, demography and vaccination coverage information (*Garske et al., 2014*; *Gaythorpe et al., 2019*). The transmission intensity is assumed to be a static force of infection, akin to the assumption that most YF infections occur as a result of sylvatic spillover (*Garske et al., 2014*; *Gaythorpe et al., 2019*). The force of infection is assumed to be constant in each province over time and age. As such, we may model the serological status of the population in age group $u$ as the following:

$$S(\lambda, u) = 1 - (1 - \frac{\sum_{a \in u}(1 - \exp(-\lambda a))p_a}{\sum_{a \in u} p_a})(1 - \frac{\sum_{a \in u} v_a p_a}{\sum_{a \in u} p_a})$$

where $\lambda$ is the force of infection, $p_a$ the population in annual age group $a$ and $v_a$ the vaccination coverage in annual age group $a$. This provides us with estimates of force of infection in specific locations where serological surveys are available.

In order to estimate transmission intensity in areas where no serological survey data is available, we link the GLM predictions with seroprevalence estimates through a Poisson reporting process. The force of infection can be used to estimate the number of infections in any year. Thus, we may calculate the number of infections over the observation period. These will be reported with a certain probability to give the occurrence shown in the GLM. As such, we assume that the probability of at least one report in a province over the observation period, $q_i$, depends on the number of infections in the following way:

$$q_i = 1 - (1 - \rho_i)^{n_{inf,i}}$$

where $\rho_c$ is the per-country reporting factor which we relate to the GLM in the following way:

$$n_{inf,i} \ln(1 - \rho_c) = -\exp(X\beta)$$

where $X$ are the model covariates and $\beta$, the coefficients. The probability of detection can then be written in terms of the country factors, which are GLM covariates, $\beta_c$, and $b$, the baseline surveillance quality calculated from the serological survey data:

$$\ln(-\ln(1 - \rho_c)) = \beta_c + b.$$

Thus, we may transform the predictions given by the GLM of YF occurrence using the probability of detection obtained in the provinces where we have both serological studies and GLM predictions to produce FOI estimates for the entire endemic region.

## Estimation

We estimate the models of temperature suitability and YF report together within a Bayesian framework using Metropolis-Hastings Markov Chain Monte Carlo sampling with an adaptive proposal distribution (*Andrieu and Thoms, 2008*; *McKinley et al., 2014*; *Roberts and Rosenthal, 2009*; *Sherlock et al., 2015*; *Tennant and McKinley, 2019*). The likelihood contains components for the GLM of YF reports as well as the thermal response models and is given by the following:

$$\log(L) = \log(L_{GLM}) + \log(L_a) + \log(L_\rho) + \log(L_\mu),$$

where $\log(L_x)$ denotes the log likelihood of element $x$. The log likelihood for the GLM assumes that the binary YF occurrence data is Bernoulli distributed (*Garske et al., 2014*):

$$\log(L_{GLM}) = \sum_i (y_i \log(q_i) + (1 - y_i) \log(1 - q_i)), \tag{2}$$

where $y_i$ is the binary occurrence and $q_i$ is the probability of at least one YF report in province $i$.

We propagate uncertainty in the estimation of the GLM from the thermal response models as well as that from the seroprevalence into the resulting transmission intensity estimates.

The thermal response likelihoods are provided by an exponential distribution for bite rate, a Bernoulli distribution for mortality and a normal distribution for extrinsic incubation period.

The estimation, analysis and manuscript were all performed or written in R version 3.5.1, ridgeline plots were generated with packages ggplot2 and ggridges (*R Development Core Team, 2014*; *Wickham, 2016*; *Wilke, 2018*; *Garnier, 2018*).

## Future projections

In order to assess future changes in force of infection, and thus disease burden, we incorporate ensemble climate projections of temperature change and precipitation. We assume that the force of infection is constant until 2018 and then changes linearly between 2018, 2050 and 2070, the years for which climate projections are available. Furthermore, in order to compare only the influence of changing population and force of infection, we assume that vaccination after 2019 is kept at the routine levels of 2018. As such, the results will not be affected by country-specific preventive vaccination campaigns but, future burden will be over estimated as there are likely to be preventive and reactive campaigns in future. We estimate burden by calculating the proportion of infections who become severe cases and then, of those, the proportions that die, using published case fatality ratio estimates (*Johansson et al., 2014*). We compare burden estimates with a baseline scenario assuming the same demographic conditions and vaccination levels as the climate change scenarios but no change in climate variables (precipitation and temperature) over time.

## Additional information

### Competing interests

Neil M Ferguson: Senior editor, eLife. The other authors declare that no competing interests exist.

### Funding

| Funder | Grant reference number | Author |
|---|---|---|
| Bill and Melinda Gates Foundation | OPP1117543 | Tini Garske<br>Katy AM Gaythorpe |
| Bill and Melinda Gates Foundation | OPP1157270 | Katy A M Gaythorpe<br>Tini Garske |
| Medical Research Council | MR/R015600/1 | Katy A M Gaythorpe<br>Arran Hamlet<br>Tini Garske<br>Neil M Ferguson |

This work was carried out as part of the Vaccine Impact Modelling Consortium (), which is funded by Gavi, the Vaccine Alliance and the Bill & Melinda Gates Foundation. The views expressed are those of the authors and not necessarily those of the Consortium or its funders. The final decision on the content of the publication was taken by the authors.We acknowledge joint Centre funding from the UK Medical Research Council and Department for International Development. The funders had no role in study design, data collection and interpretation, or the decision to submit the work for publication.

### Author contributions

Katy AM Gaythorpe, Conceptualization, Resources, Data curation, Software, Formal analysis, Validation, Investigation, Visualization, Methodology, Writing - original draft, Project administration, Writing - review and editing; Arran Hamlet, Formal analysis, Validation, Investigation, Visualization, Methodology, Writing - review and editing; Laurence Cibrelus, Data curation, Writing - review and editing; Tini Garske, Resources, Supervision, Funding acquisition, Methodology, Project administration; Neil M Ferguson, Supervision, Project administration, Writing - review and editing

## Author ORCIDs

Katy AM Gaythorpe (iD) https://orcid.org/0000-0003-3734-9081

## Decision letter and Author response

Decision letter https://doi.org/10.7554/eLife.55619.sa1
Author response https://doi.org/10.7554/eLife.55619.sa2

## Additional files

### Supplementary files

• Transparent reporting form

### Data availability

Public repository data: Vaccination coverage: coverage is available to download from the PoLiCi shiny app : https://shiny.dide.imperial.ac.uk/polici/. Serology surveys: There are seven published surveys used, available at DOI: 10.1016/0147-9571(90)90521-T, DOI: 10.1093/trstmh/tru086, DOI: 10.1186/s12889-018-5726-9, DOI: 10.4269/ajtmh.2006.74.1078, PMID: 3501739, PMID: 4004378, PMID: 3731366 Demographic data: Population level data was obtained from UN WPP https://population.un.org/wpp/, this was disaggregated using Landscan 2014 data https://landscan.ornl.gov/landscan-data-availability. Environmental data: This was obtained from LP DAAC: https://lpdaac.usgs.gov/ and worldclim http://www.worldclim.org/ Yellow fever outbreaks: These were compiled from the WHO weekly epidemiologic record and disease outbreak news https://www.who.int/wer/en/ and https://www.who.int/csr/don/en/. Data elsewhere: The data from the WHO YF surveillance database and from recent serological surveys from WHO member states in Africa underlying the results presented in the study are available from World Health Organization (contact: William Perea, pereaw@who.int or Laurence Cibrelus, cibrelusl@who.int or Jennifer Horton, jhorton@who.int).

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

# Appendix 1

## Serological surveys

The included surveys are the same as described in Gaythorpe, et al. Summary statistics for these surveys are included in the following table *Appendix 1—table 1*.

**Appendix 1—table 1.** Characteristics of included serological surveys. Recreated from *Gaythorpe et al., 2019*.

| Location | Sample size | Year | Reference |
|---|---|---|---|
| Nigeria | 184 | 1990 | *Omilabu et al., 1990* |
| Democratic Republic of the Congo | 140 | 1985 | *Werner and Huber, 1984* |
| Republic of the Congo | 360 | 1985 | *Merlin et al., 1986* |
| Cameroon (North) | 840 | 1987 | *Tsai et al., 1987* |
| Cameroon (South) | 256 | 2001 | *Kuniholm et al., 2006* |
| Uganda (zones) | 584 | 2012 | |
| Rwanda (zones) | 1286 | 2012 | |
| Zambia (zones) | 3679 | 2013 | |
| Sudan (zones) | 1814 | 2012 | |
| Kenya (zones) | 1960 | 2013 | |
| Ethiopia (zones) | 1645 | 2014 | *Mengesha Tsegaye et al., 2018* |
| Democratic republic of the Congo (zones) | 479 | 2014 | |
| South Sudan (zones) | 1480 | 2014 | |
| Chad (zones) | 352 | 2014 | |

**Appendix 1—table 2.** Parameter estimates with low and high ends of the 95% credible interval.

| Parameter | 95% CrI low | Median | 95% CrI high | Meaning |
|---|---|---|---|---|
| a_c | 0.0002 | 0.0003 | 0.0003 | Bite rate |
| a_T0 | 0.2205 | 2.9285 | 7.2004 | Bite rate |
| a_Tm | 40.0223 | 40.1368 | 40.2981 | Bite rate |
| adm05AGO | 1.1382 | 1.7656 | 2.3960 | GLM coefficients |
| adm05BDI | −1.1566 | −0.3275 | 0.4671 | GLM coefficients |
| adm05ERI | −0.9901 | −0.1074 | 0.7519 | GLM coefficients |
| adm05ETH | −1.2878 | −0.5366 | 0.1882 | GLM coefficients |
| adm05GNB | −1.4959 | −0.7566 | −0.0692 | GLM coefficients |
| adm05KEN | −1.1264 | −0.3510 | 0.3722 | GLM coefficients |
| adm05MRT | −1.1837 | −0.4218 | 0.2927 | GLM coefficients |
| adm05RWA | −1.1411 | −0.3175 | 0.4826 | GLM coefficients |
| adm05SDN | −0.8870 | −0.1106 | 0.6377 | GLM coefficients |
| adm05SOM | −1.0177 | −0.1425 | 0.7144 | GLM coefficients |
| adm05SSD | −0.9086 | −0.0796 | 0.7140 | GLM coefficients |
| adm05TZA | −1.3990 | −0.6442 | 0.0812 | GLM coefficients |
| adm05UGA | −0.6618 | −0.0081 | 0.6163 | GLM coefficients |
| adm05ZMB | −1.2049 | −0.3840 | 0.3975 | GLM coefficients |
| Intercept | −16.4268 | −13.2731 | −10.2753 | GLM coefficients |
| log.surv.qual.adm0 | 0.3209 | 0.5048 | 0.6917 | GLM coefficients |

*Appendix 1—table 2 continued on next page*

*Appendix 1—table 2 continued*

| Parameter | 95% CrI low | Median | 95% CrI high | Meaning |
|---|---|---|---|---|
| logpop | 0.9133 | 1.1466 | 1.3913 | GLM coefficients |
| lon | −1.1806 | −0.9173 | −0.6557 | GLM coefficients |
| mu_c | −0.8003 | −0.7578 | −0.7166 | Mortality |
| mu_T0 | 12.2133 | 12.7137 | 13.1498 | Mortality |
| mu_Tm | 38.0341 | 38.0481 | 38.0532 | Mortality |
| iEIP_c | 0.0001 | 0.0001 | 0.0002 | Inverse EIP |
| iEIP_T0 | 10.9412 | 17.6724 | 22.2418 | Inverse EIP |
| iEIP_Tm | 39.0737 | 42.1075 | 45.5927 | Inverse EIP |
| temp_suitability | 0.0101 | 0.1523 | 0.3863 | GLM coefficients |
| worldclim_rainfall | 0.2338 | 0.4629 | 0.6969 | GLM coefficients |
| worldclim_temp_range | −0.1912 | 0.0368 | 0.2687 | GLM coefficients |

**Appendix 1—table 3.** Deaths in the African endemic region in 2050 and 2070 compared to the baseline/constant scenario.

| Year | Scenario | Median | 95% Cr interval |
|---|---|---|---|
| 2050 | RCP 2.6 | 191309 | [62462, 468985] |
| 2050 | RCP 4.5 | 200470 | [66330, 499615] |
| 2050 | RCP 6.0 | 198096 | [65113, 489494] |
| 2050 | RCP 8.5 | 214427 | [69699, 554842] |
| 2050 | baseline | 172668 | [58177, 395300] |
| 2070 | RCP 2.6 | 273582 | [90275, 653145] |
| 2070 | RCP 4.5 | 298822 | [99694, 735109] |
| 2070 | RCP 6.0 | 301001 | [99551, 739950] |
| 2070 | RCP 8.5 | 349157 | [108913, 933389] |
| 2070 | baseline | 249556 | [84877, 560186] |

**Appendix 1—table 4.** Probability of increase (%) in deaths in the African endemic region in 2050 and 2070 compared to the baseline/constant scenario.

| Year | Scenario | Median | 95% Cr interval |
|---|---|---|---|
| 2050 | RCP 2.6 | 92.97 | [92.7, 93.23] |
| 2050 | RCP 4.5 | 94.85 | [94.61, 95.07] |
| 2050 | RCP 6.0 | 94.89 | [94.64, 95.13] |
| 2050 | RCP 8.5 | 95.98 | [95.78, 96.17] |
| 2070 | RCP 2.6 | 95.47 | [95.24, 95.7] |
| 2070 | RCP 4.5 | 94.64 | [94.41, 94.88] |
| 2070 | RCP 6.0 | 94.10 | [93.85, 94.35] |
| 2070 | RCP 8.5 | 95.94 | [95.72, 96.15] |

We estimated the components of the model within a Bayesian framework. The posterior distributions of our parameters could not be written in closed form and so we sample using Markov Chain Monte Carlo (MCMC) Materials and methods. We utilise the classical Metropolis-Hasting algorithm for sampling, where the algorithm is well described by Tennant, Mckinley and Recker. Similarly, we sample new parameters in the MCMC according to a multivariate normal distribution which is adapted according to the covariance of the Markov chain .

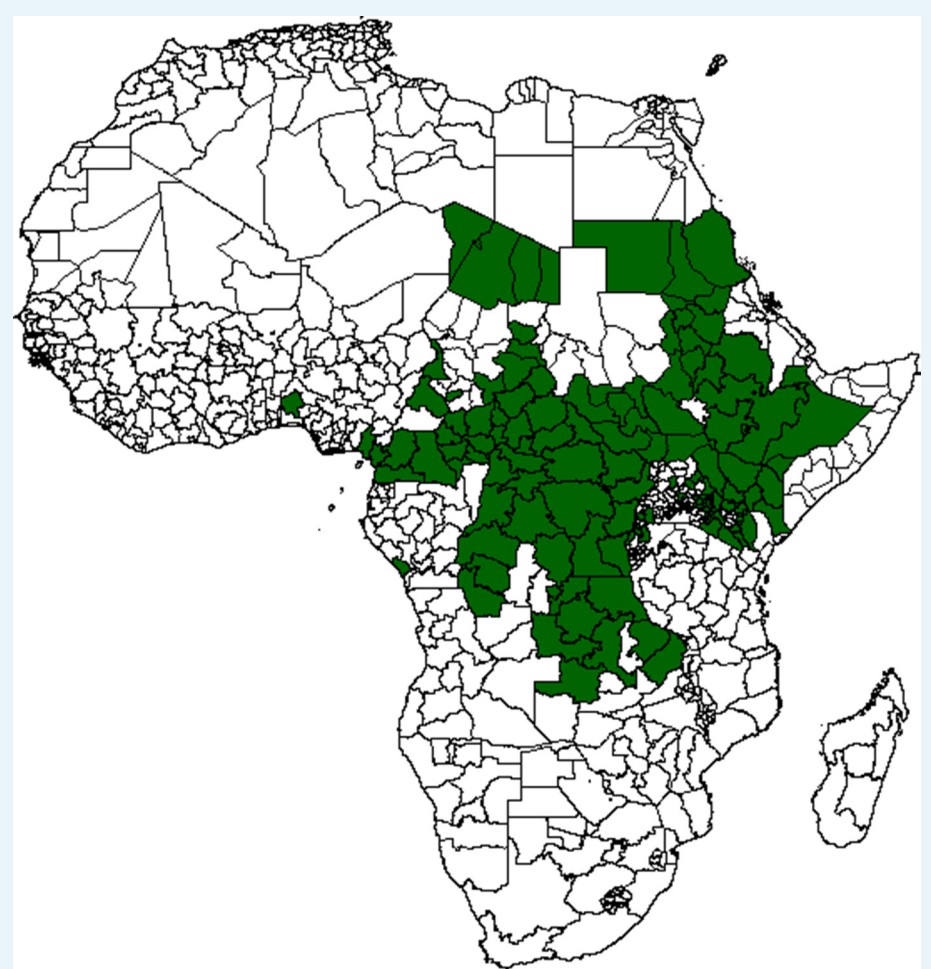

**Appendix 1—figure 1.** Location of serological study sites shown in green.

Convergence of the Markov chains was assessed visually; however, the approximate number of required iterations to achieve a standard degree of accuracy was calculated using the Raftery statistic. Trace plots of all parameters are presented in *Appendix 1—figure 2*, resulting parameter estimates are given in *Appendix 1—table 2*.

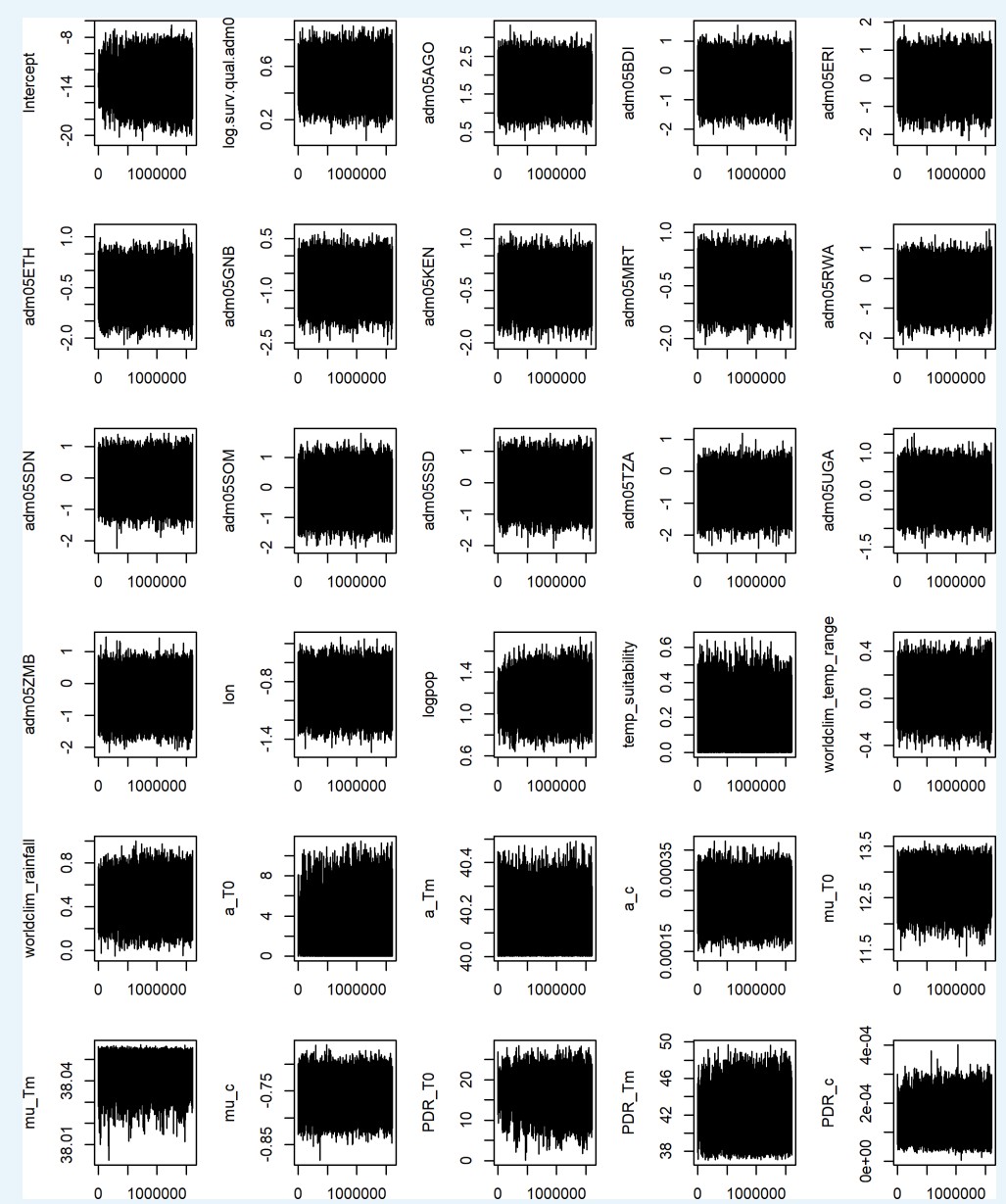

**Appendix 1—figure 2.** Traceplots for all parameters.

Twenty chains were run for between 150,000 and 260,000 iterations.

## Generalised linear model uncertainty

Uncertainty in the predictions of the generalised linear model of yellow fever risk are presented for the baseline/current scenario in *Appendix 1—figure 3*. In these, 1000 samples of the posterior predictive distribution were taken and the coefficient of variation was calculated. Uncertainty is extensive, most particularly in East and Central Africa where there are fewer reports of yellow fever occurrence. This uncertainty is propagated into the projections of risk and transmission intensity.

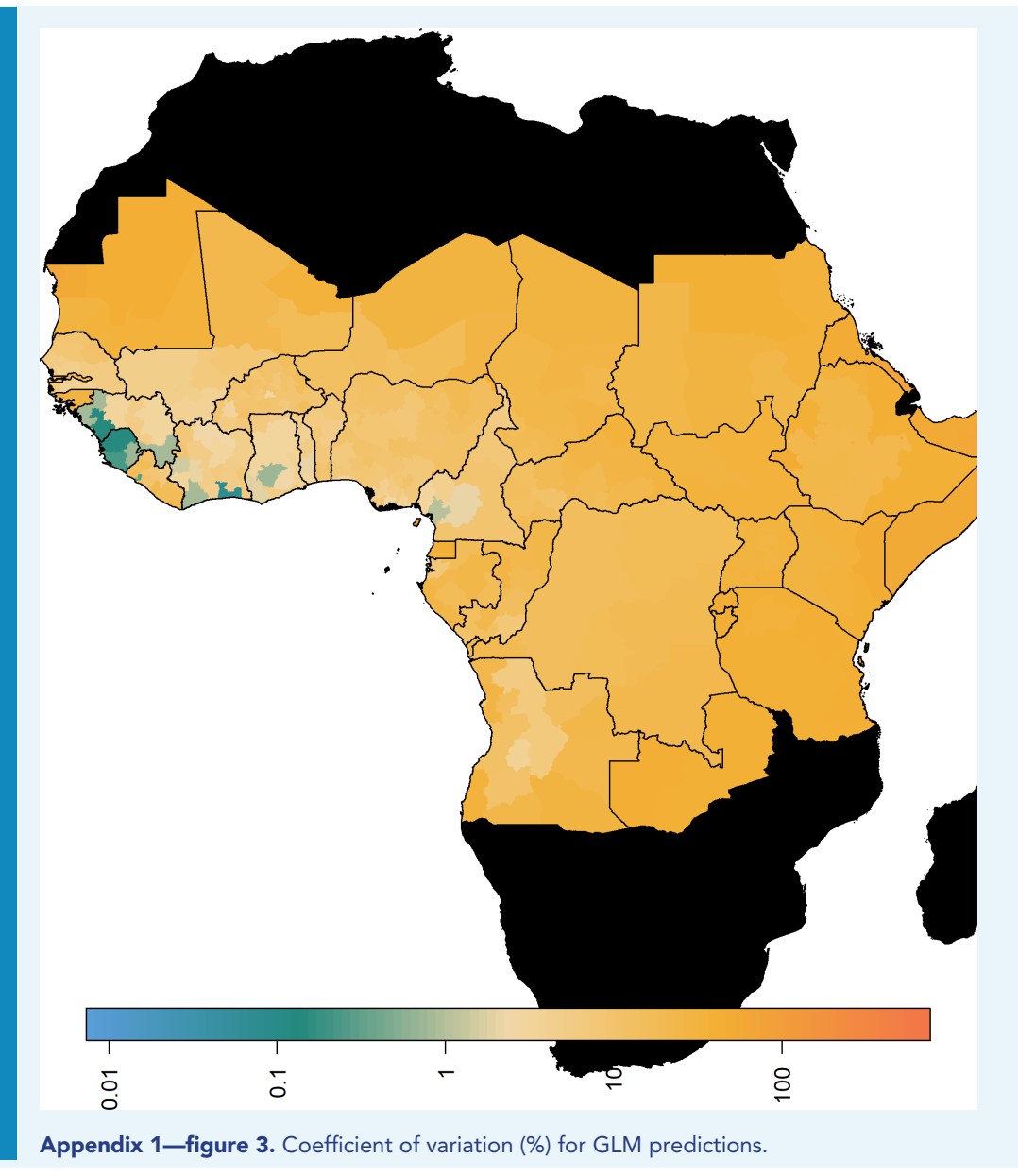

**Appendix 1—figure 3.** Coefficient of variation (%) for GLM predictions.

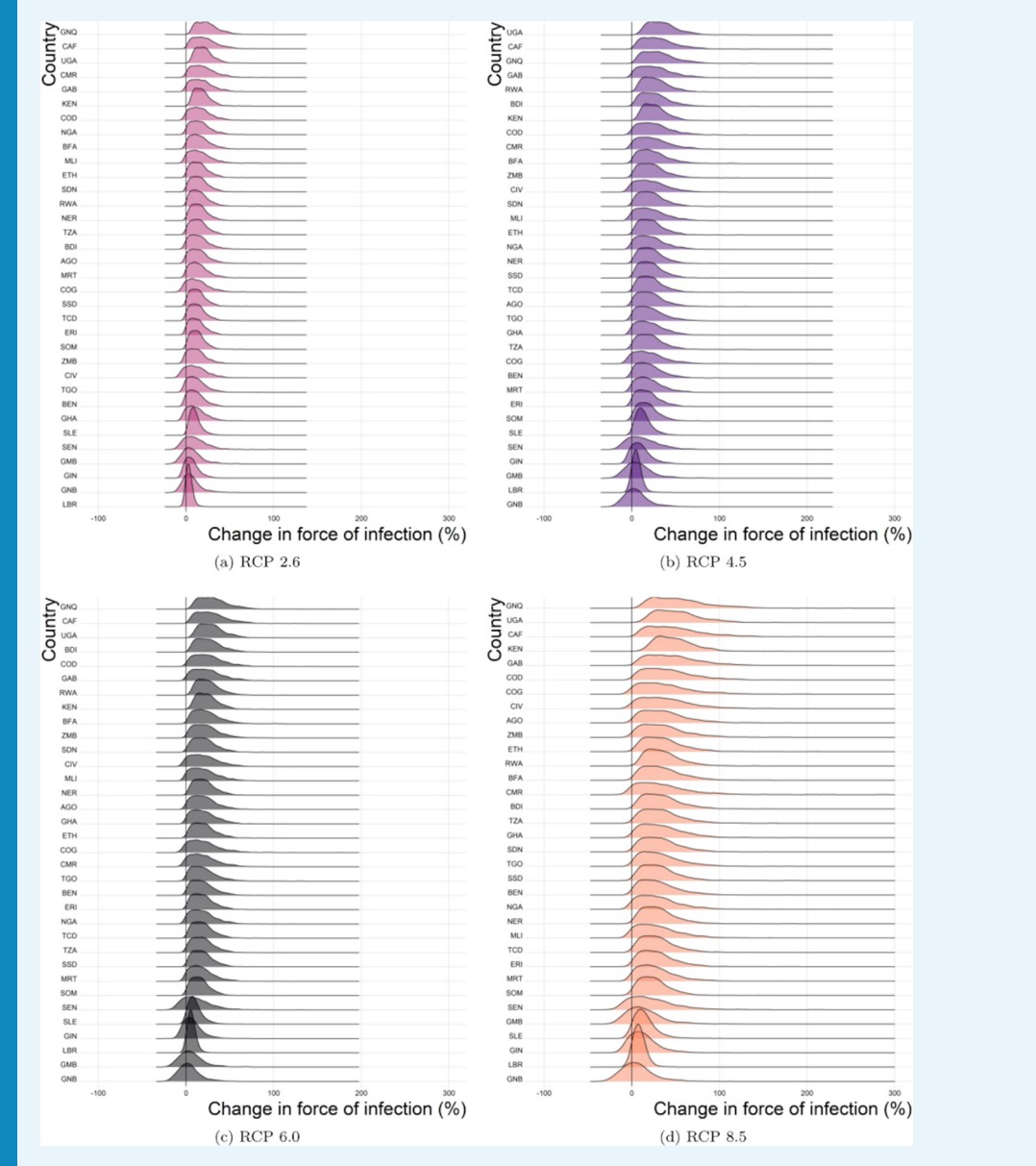

**Appendix 1—figure 4.** Predicted force of infection change (%) for each country and scenario in 2050. Note, countries are ordered by difference and may vary in position. (**a**) RCP 2.6, (**b**) RCP 4.5, (**c**) RCP 6.0, (**d**) RCP 8.5.

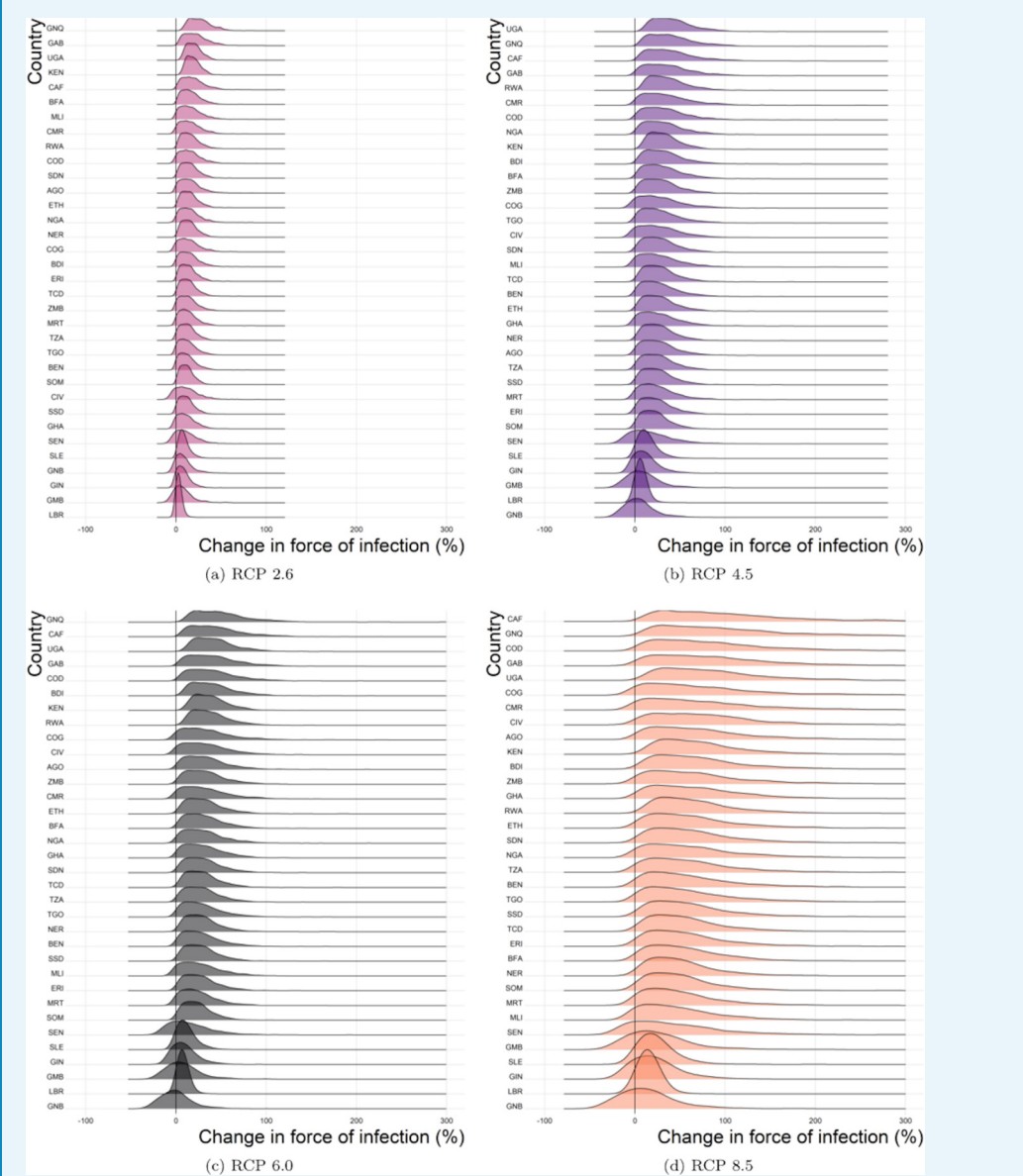

**Appendix 1—figure 5.** Predicted force of infection change (%) for each country and scenario in 2070. Note, countries are ordered by difference and may vary in position. (**a**) RCP 2.6, (**b**) RCP 4.5, (**c**) RCP 6.0, (**d**) RCP 8.5.

## Transmission intensity projection uncertainty

We include uncertainty by sampling from the collected posterior distributions 1000 times to give the results shown. This includes uncertainty both in estimated parameters and in the estimates of CFR and proportion of infections that as categorised as severe. We do not include the uncertainty in climate change projections of temperature and precipitation, as such, the uncertainty shown is an under-estimate.

The difference between scenarios is calculated between corresponding samples from the posterior predictive distribution. This means that any difference shown between scenarios is only for one parameter set.

## The influence of rainfall and/or temperature change on transmission intensity

In the following we calculate the mean transmission intensity from the median posterior predicted force of infection for each province under three scenarios:

1. That both rainfall and tempature change as per climate scenario RCP 8.5,
2. That only rainfall changes as per climate scenario RCP 8.5; temperature remains at current levels,
3. That only temeperature changes as per climate scenario RCP 8.5; rainfall remains at current levels.

In *Appendix 1—figure 6* we find heterogeneous effects by location. In general, transmission is projected to be higher when modelled with both temperature and rainfall change, this is driven by the change in temperature. In some countries such as Cameroon and Congo, including the projected change in rainfall, moderates the influence of changing temperature that is the highest transmission is seen when only temperature changes. In slight contrast, for Uganda and Kenya, the highest transmission is seen when both temperature and rainfall are changing suggesting a 'perfect storm' of climatic change.

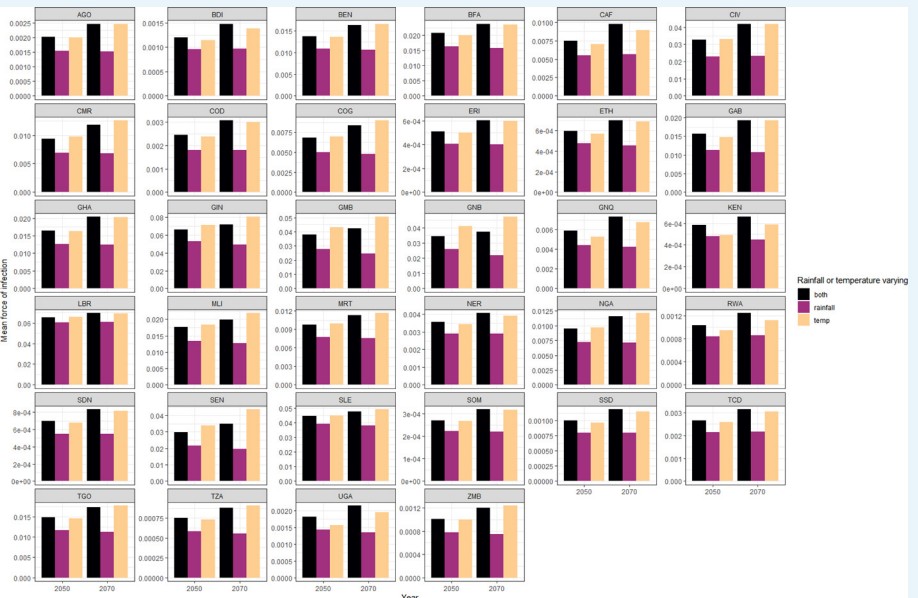

**Appendix 1—figure 6.** Predicted mean force of infection per country for 2050 and 2070 under 3 modelling scenarios. Projections are calculated from the median posterior predicted force of infection per province. All projections assume temperature and/or rainfall changing under RCP 8.5.

*Appendix 1—figure 6* is included for illustration of the relative effects of changes in rainfall and temperature across the region. In all climate scenarios, both rainfall and temperature are projected to change (see earlier figures) and so for the main text, we include the influence of both for all projections. projected-number-of-deaths.

## Projected number of deaths

The projected numbers of deaths in the African endemic region for 2050 and 2070 are shown in *Appendix 1—table 3.* and the probability of an increase in deaths is shown in *Appendix 1—table 4.*

