## [Decision Letter]

**Acceptance summary:**

This is an important piece of work, of both clear public health importance and of scientific interest, that will guide policies and programmes against Yellow Fever across sub-Saharan Africa in the years ahead due to the potential impact of climate change on the intensity and distribution of yellow fever transmission and disease burden there.

**Decision letter after peer review:**

Thank you for submitting your article "The effect of climate change on Yellow Fever disease burden in Africa" for consideration by *eLife*. Your article has been reviewed by three peer reviewers, one of whom is a member of our Board of Reviewing Editors, and the evaluation has been overseen by Eduardo Franco as the Senior Editor. The reviewers have opted to remain anonymous.

The reviewers have discussed the reviews with one another and the Reviewing Editor has drafted this decision to help you prepare a revised submission.

Summary:

The reviewers agreed that this is an important piece of work – of both clear public health importance and of scientific interest – that will guide policies and programmes against Yellow Fever across sub-Saharan Africa in the years ahead due to the potential impact of climate change on the intensity and distribution of yellow fever transmission and disease burden there.

Essential revisions:

1) The description of approach would benefit greatly from a relatively simple schematic diagram to make it very clear how the different data (seroprevalence; occurrence etc.) and modelling components (temperature suitability model; seroprevalence/transmission/FOI model; covariate-driven GLM) link together.

2) Framework for inferring contemporary pattern of transmission and burden

- The authors should please provide further details on how the YF occurrence database was compiled and contents of the database. For example, what was the inclusion/exclusion criteria for an occurrence? Was potential cross-reactivity with other flaviviruses considered? For data extracted from WERs, DONs and literature – at the very least a reference list should be provided.

- Can you present a map showing areas where you infer FOI based on serological surveys and where it is inferred as a function of the GLM 'probability of a report' estimate.

- Some more narrative would also be helpful in explaining conceptually how the FOI-to-probability of occurrence component was achieved. It appears that the ingredients you have in each province for which no sero survey exists are (i) the GLM-derived 'probability of > = 1 case in a given province'; (ii) population; (iii) an idea of the probability that a case is detected given that it occurs. Your objective is to infer FOI. It is not clear if you do this within your Bayesian inference framework-thereby somehow borrowing strength from those provinces for which you have both sero surveys and GLM estimates, or, if this is essentially an arithmetic transform that converts the GLM output to an FOI estimate. It appears to be the latter-and if so it is then not clear how robust this transform is, how you infer probability of detection etc. Presumably you could also report on the degree of alignment between your GLM outputs and you FOI calculations in provinces that do have sero-surveys, as an indicator of how well this approach will then work in provinces without surveys.

- On YF burden from zoonotic/intermediate transmission versus urban transmission cycles ("the transmission intensity is assumed to be a static force of infection, akin to the assumption that most YF infections occur as a result of sylvatic spillover." And also in the Discussion.) Instead of only touching on it in the Discussion perhaps it should be highlighted that the estimate of FOI is limited to burden from zoonotic spillover earlier/throughout the manuscript. Consider describing results as estimated effect of climate change on zoonotic/sylvatic YF burden?

- How appropriate is it to use temperature suitability pertaining only to the urban vector rather than vectors involved in the intermediate and sylvatic transmission cycles. Also, whether occurrences from urban outbreaks should be excluded from the occurrence database?

- Can you clarify also what is the implication of focusing on *Ae. aegypti* (subsection “Temperature suitability”) instead of other vectors?

3) Framework for inferring contemporary relationship with rainfall and temperature

- There is a much greater emphasis/level of sophistication around temperature effects than around rainfall. For temperature, the authors recognise the highly non-linear response via a mechanistic model capturing deterministically the temperature effects on different components of the transmission cycle. For rainfall, however, the presumably similarly complex mechanisms of impact on aedes breeding, vector densities, interactions with humans etc., are represented much more crudely via a single covariate in a GLM, with implicit assumptions around linearity. Some justification on this would be useful.

- While the paper builds on previous work, the very small number of covariates in the GLM is surprising and needs justification in this current paper. Is there no rationale for including, for example, human population density, or metrics of landscape proxies for non-human primate habitats etc?

- Further, could there be a bit more discussion of the reasonableness and motivation for omitting land cover and vegetation index? This would seem to be important given that at least in Brazil there's a very strong effect of proximity to forest?

4) Framework for inferring impact of changing climate

- Given it would be straightforward in your current framework, please consider the utility of running projections with *only* the effects of temperature change and *only* the effects of rainfall change? The rationale being that you have differing levels of sophistication in your ability to link these to YF risk; that there are differing levels of certainty in the climate projections for these two aspects; and that you could provide a richer interpretation of their respective contributions to elevated risk.

- You elude to it in the Discussion, but more consideration of what you do *not* include in your framework is crucial here. Temperature/rainfall changes may exert upward pressure on transmission and burden – but how does the magnitude of this effect compare to, for example, a modest increase in vaccination coverage, or changes in exposure due to urbanisation or modernisation of housing etc. How might global change impact other components of the transmission cycle such as the col-location of humans an NHP zoonotic reservoir species?

5) Impact of vaccination

- It is not clear from this paper or the references precisely what vaccination coverage data was compiled and how it was used in vaccination coverage estimates. The data and methods are not fully described in this paper or the referenced papers. It seems that uncertainty in vaccination coverage estimates was considered in Garske et al., 2014, by exploring alternate vaccination targeting assumptions. For the analysis in this manuscript, it appears that only the scenario where supplemental vaccination campaigns target unvaccinated individuals is considered? Lessler and colleagues (journal.pmed.1001110) have previously described issues with assumptions about vaccination targeting when making coverage estimates. It is currently not clear whether any uncertainties were propagated through the analyses presented in this paper. It is important to explore the possible impact of these (potentially large) uncertainties on the results.

6) Impact of future population changes

- Why were demographic conditions held constant? There ought to be projections for future population growth (and ageing?) for these countries that would give a more accurate representation of future disease burden.

- Possible to forecast total numbers of deaths as well as the country specific ones?

---

## [Author Response]

Essential revisions:1) The description of approach would benefit greatly from a relatively simple schematic diagram to make it very clear how the different data (seroprevalence; occurrence etc.) and modelling components (temperature suitability model; seroprevalence/transmission/FOI model; covariate-driven GLM) link together.

Thank you, we have included a schematic of this information (Figure 1—figure supplement 1).

2) Framework for inferring contemporary pattern of transmission and burden- The authors should please provide further details on how the YF occurrence database was compiled and contents of the database. For example, what was the inclusion/exclusion criteria for an occurrence? Was potential cross-reactivity with other flaviviruses considered? For data extracted from WERs, DONs and literature – at the very least a reference list should be provided.

We have included further details in the methodology to expand on how this was collated (section 2.1.1). This database was detailed first in Garske et al., 2013, and has been updated since. Cross reactivity with other flaviviruses was not included.

- Can you present a map showing areas where you infer FOI based on serological surveys and where it is inferred as a function of the GLM 'probability of a report' estimate.

Thank you, a map of serological study locations has been included in the Appendix (Appendix—figure 1).

- Some more narrative would also be helpful in explaining conceptually how the FOI-to-probability of occurrence component was achieved. It appears that the ingredients you have in each province for which no sero survey exists are (i) the GLM-derived 'probability of > = 1 case in a given province'; (ii) population; (iii) an idea of the probability that a case is detected given that it occurs. Your objective is to infer FOI. It is not clear if you do this within your Bayesian inference framework-thereby somehow borrowing strength from those provinces for which you have both sero surveys and GLM estimates, or, if this is essentially an arithmetic transform that converts the GLM output to an FOI estimate. It appears to be the latter-and if so it is then not clear how robust this transform is, how you infer probability of detection etc. Presumably you could also report on the degree of alignment between your GLM outputs and you FOI calculations in provinces that do have sero-surveys, as an indicator of how well this approach will then work in provinces without surveys.

We have added further description to clarify the transform (subsection “Mapping probability of occurrence to force of infection”) and added a comparison in the Appendix for the alignment between the FOI from serology and from the GLM.

- On YF burden from zoonotic/intermediate transmission versus urban transmission cycles ("the transmission intensity is assumed to be a static force of infection, akin to the assumption that most YF infections occur as a result of sylvatic spillover." And also in the Discussion.) Instead of only touching on it in the Discussion perhaps it should be highlighted that the estimate of FOI is limited to burden from zoonotic spillover earlier/throughout the manuscript. Consider describing results as estimated effect of climate change on zoonotic/sylvatic YF burden?

Thank you, we have reviewed the text and included a statement at the beginning of the Results section to highlight this.

- How appropriate is it to use temperature suitability pertaining only to the urban vector rather than vectors involved in the intermediate and sylvatic transmission cycles. Also, whether occurrences from urban outbreaks should be excluded from the occurrence database?

Urban outbreaks result first from a spillover event and the majority of the outbreaks are considered to be sylvatic in Africa, as such it would not be necessary to exclude further data form the database.

Using information on the urban vector is an approximation made due to data limitations. However, *Aedes aegypti* is well examined and there are studies available investigating the thermal response of them in different contexts. We also fit the temperature dependence in a hierarchical framework in order to capture some of the uncertainty resulting from the lack of sylvatic vector data. We have added further information in the description of data sources and Discussion to describe this limitation.

- Can you clarify also what is the implication of focusing on Ae. aegypti (subsection “Temperature suitability”) instead of other vectors?

As we focus on one vector rather than the multitude that are related to YF transmission, it is likely that we have underestimated uncertainty in the response of the vector to changes in climate. We have added to the Discussion to highlight this.

3) Framework for inferring contemporary relationship with rainfall and temperature- There is a much greater emphasis/level of sophistication around temperature effects than around rainfall. For temperature, the authors recognise the highly non-linear response via a mechanistic model capturing deterministically the temperature effects on different components of the transmission cycle. For rainfall, however, the presumably similarly complex mechanisms of impact on aedes breeding, vector densities, interactions with humans etc., are represented much more crudely via a single covariate in a GLM, with implicit assumptions around linearity. Some justification on this would be useful.

The relationship between the thermal response of vector attributes such as mortality are well characterised in comparison the influence of rainfall. We have added further description on this in the Discussion.

- While the paper builds on previous work, the very small number of covariates in the GLM is surprising and needs justification in this current paper. Is there no rationale for including, for example, human population density, or metrics of landscape proxies for non-human primate habitats etc?

We include human population density – this was accidentally omitted from the table in the Appendix which is now corrected (Appendix —table 2).

Human population is included and the landcover metrics are omitted due to uncertainties oh how they will be affected by climate change. Similarly, whilst we use projections of human population sizes which include measures of immigration, the distribution of humans and non-human primates are uncertain and are likely to vary in the future and as a product of climate change. We have added to the description of this in the Discussion.

- Further, could there be a bit more discussion of the reasonableness and motivation for omitting land cover and vegetation index? This would seem to be important given that at least in Brazil there's a very strong effect of proximity to forest?

Thank you. This limitation is detailed in the Discussion. We required a suite of covariates that both reflected the transmission now but also could be parametrised for the 4 climate scenarios. The vegetation index is thus far not included for this reason.

4) Framework for inferring impact of changing climate-Given it would be straightforward in your current framework, please consider the utility of running projections with only the effects of temperature change and only the effects of rainfall change? The rationale being that you have differing levels of sophistication in your ability to link these to YF risk; that there are differing levels of certainty in the climate projections for these two aspects; and that you could provide a richer interpretation of their respective contributions to elevated risk.

Thank you, this was an interesting point. We have included a further subsection, “Environmental and climate projections”, where we show the change to the mean force of infection per country in 2050 and 2070 given either rainfall *or* temperature changes under climate scenario RCP 8.5. We find that the dominant factor is the change in temperature but that rainfall and temperature change can have interesting effects: in some cases, the change in rainfall mitigates the influence of changing temperature, in others, such as Uganda, the change in both creates a ‘perfect storm’ where transmission is projected to be highest if both change under the projected scenario.

We have added a further section to the Discussion for this point.

- You elude to it in the Discussion, but more consideration of what you do not include in your framework is crucial here. Temperature/rainfall changes may exert upward pressure on transmission and burden – but how does the magnitude of this effect compare to, for example, a modest increase in vaccination coverage, or changes in exposure due to urbanisation or modernisation of housing etc. How might global change impact other components of the transmission cycle such as the col-location of humans an NHP zoonotic reservoir species?

Thank you, we have expanded the Discussion to further highlight these limitations and that, as this time, projecting these highly non-linear influences may not be possible with available data.

5) Impact of vaccination- It is not clear from this paper or the references precisely what vaccination coverage data was compiled and how it was used in vaccination coverage estimates. The data and methods are not fully described in this paper or the referenced papers. It seems that uncertainty in vaccination coverage estimates was considered in Garske et al., 2014, by exploring alternate vaccination targeting assumptions. For the analysis in this manuscript, it appears that only the scenario where supplemental vaccination campaigns target unvaccinated individuals is considered? Lessler and colleagues (journal.pmed.1001110) have previously described issues with assumptions about vaccination targeting when making coverage estimates. It is currently not clear whether any uncertainties were propagated through the analyses presented in this paper. It is important to explore the possible impact of these (potentially large) uncertainties on the results.

We have included further details in how the vaccination coverage was calculated and highlighted the limitations around quantifying uncertainty in vaccination and subsection “Past vaccination coverage and demography”, Discussion). We have also emphasised that this is based on methodology that is available in an R package, as a shiny application and has been detailed in both the POLICI application paper [Hamlet et al., 2019] including how campaigns are targeted, and in [Garske et al., 2014]. In this manuscript, vaccination campaigns are assumed to target people at random rather than only unvaccinated individuals. We also focus on using the same vaccination scenarios in all projections and do not specifically quantify vaccine impact here, although, as noted in the Discussion, it may mitigate some of the effects of changing transmission.

6) Impact of future population changes- Why were demographic conditions held constant? There ought to be projections for future population growth (and ageing?) for these countries that would give a more accurate representation of future disease burden.

Demographic conditions are assumed to follow the United Nations World Population Prospects (UNWPP) projections of population growth. As such, demographic conditions are not held constant and future projections of burden take these into account. The counterfactual also assumes population growth.

- Possible to forecast total numbers of deaths as well as the country specific ones?

Sure, we have added a table to the Appendix for 2050 and 2070 for all scenarios (Appendix—table 4).